# Efficacy and Safety of Fecal Microbiota Transplantation in Treatment of *Clostridioides difficile* Infection among Pediatric Patients: A Systematic Review and Meta-Analysis

**DOI:** 10.3390/microorganisms10122450

**Published:** 2022-12-12

**Authors:** Kyaw Min Tun, Mark Hsu, Kavita Batra, Chun-Han Lo, Tooba Laeeq, Tahne Vongsavath, Salman Mohammed, Annie S. Hong

**Affiliations:** 1Department of Internal Medicine, Kirk Kerkorian School of Medicine at UNLV, University of Nevada, Las Vegas, NV 89102, USA; 2Department of Medical Education, Kirk Kerkorian School of Medicine at UNLV, University of Nevada, Las Vegas, NV 89102, USA; 3Office of Research, Kirk Kerkorian School of Medicine at UNLV, University of Nevada, Las Vegas, NV 89102, USA; 4Division of Gastroenterology and Hepatology, Department of Internal Medicine, Kirk Kerkorian School of Medicine at UNLV, University of Nevada, Las Vegas, NV 89102, USA

**Keywords:** pediatrics, *Clostridioides difficile* infection, fecal microbiota transplantation, diarrhea

## Abstract

Background and Aims: Cases of *Clostridioides difficile* infection have been rising among the pediatric and adolescent population. Fecal microbiota transplantation (FMT) has emerged as an alternative therapy for recurrent *C. difficile* infection. We aim to perform the first systematic review and meta-analysis investigating the safety and efficacy of fecal microbiota transplantation for *C. difficile* infection in children and adolescents. Methods: A literature search was performed using variations of the keywords “pediatrics”, “*C. difficile* infection”, and “fecal microbiota transplantation” in PubMed, EMBASE, CINAHL, Cochrane, and Google Scholar from inception to 30 June 2022. The resulting 575 articles were independently screened by three authors. Fourteen studies that satisfied the eligibility criteria were included in the meta-analysis. Results: The pooled success rate of FMT in the overall cohort was 86% (95% confidence interval: 77–95%; *p* < 0.001; I^2^ = 70%). There were 38 serious adverse events in 36 patients with a pooled rate of 2.0% (95% confidence interval: 0.0–3.0%; *p* = 0.1; I^2^ = 0.0%) and 47 adverse events in 45 patients with a pooled rate of 15% (95% confidence interval: 5.0–25.0%; *p* = 0.02; I^2^ = 54.0%). There was no death associated with FMT. Conclusions: FMT was concluded to be an effective and safe therapy in pediatric and adolescent patients with *C. difficile* infection. Underlying comorbidities may impede the efficacy. A rigorous screening process of the donors is recommended prior to embarking on FMT. There is no universal and cost-effective way to monitor the long-term outcomes of FMT. While promising, metagenomic sequencing may not be available in settings with limited resources. Robust data from randomized clinical trials is warranted.

## 1. Introduction

*Clostridioides difficile* (formerly *Clostridium difficile*) is a nosocomial infection that causes severe watery diarrhea and is associated with increased hospitalization, healthcare costs, morbidity, and mortality [1,2,3]. In the United States (U.S.) alone, *C. difficile* infection (CDI) was responsible for half a million infections and over USD 1.5 billion in excess healthcare expenditure in 2011 [3]. Approximately 20% of patients who were diagnosed with CDI in 2011 had a recurrence of CDI, which is defined as persistence of symptoms within 60 days of completion of previous treatment [4]. Of those with recurrent CDI (rCDI), 29,000 patients died from rCDI [4]. Originally thought to affect adults only, CDI has also been increasing in its incidence among the pediatric population in both inpatient (hospital-acquired) and outpatient (community-acquired) settings over the past 20 years [4,5,6,7]. A 12.5-fold increase in incidence of CDI was observed among the pediatric population from 1991 to 2009 [5,6]. Furthermore, up to 35% of pediatric patients may experience recurrence after treatment with first-line agents [7]. The causes of recurrence include, but are not limited to dysbiosis of the microbiome, continued exposure to *C. difficile*, and immunocompromised status [5]. Although there is a substantial amount of published data available for CDI in adults, CDI in the pediatric population has been increasingly studied only in recent years. There is a paucity of information on CDI in pediatric patients compared to their adult counterparts with regard to the severity of the disease, the treatments available, the outcomes, and its impact on the pediatric patients.

The traditional treatment of CDI includes antibiotics such as fidaxomicin, oral or rectal vancomycin, and parenteral metronidazole as a single agent and/or in combination [1]. As dysbiosis and reduced biodiversity of the gut microbiome has been noted among patients with CDI or rCDI [6], fecal microbiota transplantation (FMT) has emerged in recent years as an alternative therapy for rCDI [3]. The donor sample typically originates from adults and can be attained through various preparations. For instance, the donor may be related to the recipient, such as a family member, or unrelated [3]. In either case, it is paramount that both donors and recipients undergo a screening process and have their stools analyzed for microbiome composition prior to the procedure [3,7]. The sample itself may be collected as a fresh specimen within 1–2 days prior to the transplantation or may be prepared from a frozen donor stool from the stool bank [7]. FMT can be accomplished via different routes of administration including, but not limited to, colonoscopy, capsules, and nasogastric tubes. 

In adult patients, FMT has been demonstrated to be an effective therapy against CDI or rCDI [1,5]. The first successful use of FMT in the pediatric population for CDI was reported in the literature in 2010 [5]. In addition, FMT has been studied as a form of treatment in a variety of pediatric conditions such as inflammatory bowel disease, autism spectrum, and obesity [1,2,3,4,5,6,7]. Since 2010, several studies have investigated the effectiveness of FMT for the treatment of CDI in adolescents and pediatric patients. In 2019, the North American Society for Pediatric Gastroenterology, Hepatology and Nutrition, and the European Society for Pediatric Gastroenterology, Hepatology, and Nutrition published a joint recommendation that FMT may be used in pediatric patients with rCDI [8]. However, the long-term effects (positive or negative) of alterations to the gut microbiome through FMT in the children and adolescent population remain to be seen. Therefore, it was recommended that FMT be performed only in experienced centers where the long-term effects could be monitored [8].

While FMT has been observed as an effective and safe therapy against CDI in children, adolescents, and young adults, there are currently no randomized controlled trials published in the literature and the available information is derived from observational studies, case series, and case reports [1,2,3,4,5,6,7,9,10]. Nonetheless, the current data from the literature must be studied and analyzed for better elucidation of the available information. Therefore, we aim to conduct the first systematic review and meta-analysis to determine the efficacy and safety of FMT in the treatment of CDI and/or rCDI in the patients <18 years of age, adolescents, and young adults.

## 2. Materials and Methods

### 2.1. Search Strategy

We performed a comprehensive literature search across five databases (Pubmed/Medline, EMBASE, CINAHL, Cochrane, Web of Science, and Google Scholar) using variations of the keywords “fecal microbiota transplant” and “pediatric” to identify original studies published from inception through to 30 June 2022. Results were limited to human studies published in English. There were a total of 575 studies for review. 

Prior to screening the studies for eligibility to be included in the systematic review and meta-analysis, our review was registered on PROSPERO (PROSPERO registration number CRD42022343342; Registered 30 June 2022). See Appendix A for detailed search terms.

### 2.2. Eligibility Criteria

Inclusion criteria were: (1) FMT for CDI and/or rCDI; (2) pediatric patients 21 years old or younger; (3) reporting of patient data and outcomes after first fecal infusion; (4) patients of any sex; (5) minimum follow-up of 2 months; (6) sample size of at least 5 patients; and (7) at least moderate quality of evidence. In 2017, the American Academy of Pediatrics defined adolescence from 12 to 21 years of age, and identified 21 years as the upper age limit of the pediatric population [11]. Furthermore, several FMT studies on pediatrics included age up to 21 years in the sample [5,12]. Therefore, patients of age up to 21 years old were included in our systematic review and meta-analysis.

Exclusion criteria: (1) case reports with less than 5 patients; (2) published abstracts, letters to editor, and commentaries which did not require detailed patient data or an extensive review process; (3) studies without patient data; (4) non-English studies; and (5) animal studies. Case series with more than 5 patients were included in our systematic review and meta-analysis. The threshold for the number of patients that distinguished between case series (5 or more patients) and case reports (less than 5 patients) was derived from a prior concept analysis by Abu-Zidan et al. [13].

### 2.3. Quality Assessment

Newcastle–Ottawa Scale (NOS) is used to evaluate the methodological quality in observational studies such as case-control and cohort studies. The risk of bias regarding the selection of subjects, comparability of subjects, and assessment of the exposure and outcome is graded by using a star system corresponding to nine items. A study is categorized as low risk of bias if a total of 8 to 9 stars are allocated, medium risk of bias if 6 to 7 stars are allocated, and high risk of bias if the study is given ≤5 stars [14].

For case series, the appraisal of quality and risk of bias was performed by a series of quality assessment tools developed by the US National Heart Lung and Blood Institute (NHLBI) of the National Institutes of Health (NIH, Bethesda, MD, USA) (https://www.nhlbi.nih.gov/health-topics/study-quality-assessment-tools) (accessed on 1 August 2022). Similar to the NOS, a set of question items with yes/no answers were used, with a “Yes” counting as a score of 1 and a “No” as a score of 0. In this tool used for case series, there were a total of 9 questions. A score of 7–9 corresponded to good quality, while scores of 4–6 and 1–3 indicated moderate and poor quality, respectively [14].

In the final selection stage, only studies with at least a moderate level of evidence were included. Quality appraisal was performed by at least two of the following authors (K.M.T., C.H.L., T.L., and T.V.). If there was any disagreement, a senior reviewer (A.H.) evaluated the article and achieved consensus through discussion. See Appendix A for quality assessment scores for each study.

### 2.4. Study Outcomes and Effect Size

The primary endpoint was the efficacy or clinical success of FMT in the treatment of rCDI among the pediatric patient population. The term “success” was defined as the resolution of symptoms (≥3 watery bowel movements in ≤24 h) and no requirement for further interventions for CDI for at least 8 weeks after the date of FMT [1,4]. The term “failure” or “recurrence” was used to describe an initial episode of CDI followed by persistence of symptoms or return of symptoms that needed further treatment for CDI within at least 8 weeks from the date of FMT [1,4,15]. Additionally, recurrence of CDI also required positive laboratory testing, such as a nucleic acid amplification test or stool toxin test in patients who remained symptomatic after FMT [1,15]. The effect size used in this study was the event rate (success rate). The event or success rate was calculated by dividing the number of reported events by the total sample size of the individual studies.

The secondary endpoint was the safety of FMT, which can be divided into serious adverse events (SAE) and adverse events (AE) that occurred within at least 8 weeks after FMT [15]. The SAEs and AEs were specifically labeled as attributable to FMT by the respective authors from each study. If there was no specific delineation, it was assumed that all the SAEs and AEs were related to FMT. SAEs were defined as death or any event requiring hospitalization. AEs included other adverse events that did not meet the criteria for SAE.

### 2.5. Study Selection and Data Extraction

A total of 575 articles were retrieved in the initial search. Two authors (K.M.T. and M.H.) independently reviewed these titles and abstracts. Afterwards, 23 articles were deemed relevant with patient data. Full texts were then reviewed by at least two of the following authors (K.M.T., C.H.L., T.L. and T.V.), after which 14 studies fulfilled the eligibility criteria. In cases of disagreement, a senior reviewer (A.H.) arbitrated the final decision for inclusion. The study selection process by the preferred reporting items for systematic reviews and meta-analyses (PRISMA) statement is detailed in Figure 1. A list of 20 articles (as shown in Figure 1) that were retrieved and reviewed can be found in Appendix A. IRB review was not required as all data were extracted from published literature and no patient intervention was directly performed.

### 2.6. Data Analysis

Individual estimates of each study were pooled to compute the summary estimates of the clinical success or efficacy of the FMT. A weighted summary statistic was calculated if many zero values occurred (e.g., adverse events) to prevent positive bias. The inverse variance heterogeneity (IVhet) model was fitted to account for methodological differences among the included studies for generating summary estimates [16]. The strength of evidence of heterogeneity across studies was determined by Cochran’s Q and I^2^ statistics [17,18,19]. The values of under 30, 30–60, 61–75, and over 75% were categorized as low, moderate, substantial, and considerable heterogeneity, respectively [20]). Subgroup analyses of the clinical success of FMT by gender were also performed. Sensitivity analysis was conducted to determine the validity of the estimated summary effect size. For the sensitivity analysis, a “leave-one-out analysis” was conducted to investigate the impact of the removal of study (one by one) on the estimates and LFK index asymmetry (Luis Furuya-Kanamori index). Publication bias was assessed by visually inspecting funnel plot and Doi plot [21,22]. In addition, Luis Furuya-Kanamori (LFK) index was used as a quantitative method to assess asymmetry of the study effects or publication bias as it has been noted in the literature that LFK index has higher sensitivity than the Egger regression statistics, particularly in meta-analysis with a small number of studies [21]. All meta-analyses were performed using MetaXL software (v. 5.3; EpiGear International, Sunrise Beach, Queensland, Australia). The 95% Clopper–Pearson exact confidence intervals and prediction intervals were calculated using R package [23,24].

## 3. Results

Table 1 summarizes the studies included in the systematic review and meta-analysis. There was a total of 14 studies that comprised five retrospective observational studies, five prospective observational studies, and four case series. In total, there were 904 pediatric patients who received FMT. In Nicholson, 2022 [1], the total number of patients who received FMT was 396, with success rates of 81.85% (203/248), 75.68% (112/148), and 79.55% (315/396), respectively, in the non-IBD, IBD, and overall cohorts. While efficacy was reported for the overall sample cohort, the following data was reported only for IBD cohort: gender, number of times of FMT administered, routes of FMT delivery, SAE, and AE.

There were 326 male patients and 303 female patients from the studies that reported the genders of those who received FMT. Three studies (Aldrich, 2019 [4], Hourigan, 2019 [7], and Hourigan, 2015 [18]) did not distinguish the gender among the FMT patients. Nicholson, 2022 [1] reported gender data only for the IBD cohort as discussed above. The mean age of patients was 9.38 ± 2.80 years. There were five multi-center studies and nine single-center studies. All studies were performed within the U.S.A (North America), except for Li, 2022, [2] which was conducted in China (Asia). Almost all studies performed FMT for rCDI except for Nicholson, 2020 [5] and Barfield, 2018 [25], where FMT was used for both CDI and rCDI patients.

There were a total of 725 times FMT was administered since there were patients who received FMT more than once. Table 2 demonstrates the various routes via which FMT was administered. Delivery through the upper gastrointestinal tract included esophagogastroduodenoscopy, capsule, nasogastric, nasoduodenal, nasojejunal, gastric, duodenal, or jejunostomy tubes. Delivery through the lower gastrointestinal tract included colonoscopy, sigmoidoscopy, or enema. Colonoscopy was the most frequently used technique.

Inflammatory bowel disease (IBD) was the most common concomitant disease, and was found in 337 patients (Table 3). Other concurrent gastrointestinal diseases included gastroesophageal reflux disease (*n* = 38), short bowel syndrome (*n* = 10), and celiac disease (*n* = 1). Neuromuscular disorders included epilepsy, Lennox–Gastaut syndrome, Emmanuel syndrome, and muscular dystrophy, while genetic disorders such as mitochondrial disease, and cystic fibrosis were also observed.

### 3.1. Primary Outcome

The primary outcome was the efficacy of FMT in treating CDI or rCDI. Success was defined as the resolution of diarrhea symptoms (<3 watery bowel movements in ≤24 h) without requiring further treatment or interventions for CDI for at least 8 weeks after receiving FMT. Table 4 shows the number of patients who had success and those who had failure with the treatment in the FMT cohort. The rate of success ranged between 66 and 100%, the latter of which was found in seven studies. The gross success rate was 81.86% (740/904) while the overall failure rate was 18.14% (164/904).

Table 5 further differentiates the FMT success cohort by gender. Some studies did not report data on gender and are thus labeled as not reported (NR). The gross success rate was 78.98% (233/295) in males and 82.51% (217/263) in females. The failure rates were 21.02% (62/295) and 17.49% (46/263) among male and female patients, respectively.

The calculated pooled rate of clinical success of FMT in the overall cohort was 86% (95% confidence interval [CI]: 77%, 95%; *p* < 0.001; I^2^ = 70%). Figure 2 shows the forest plot for clinical success or the efficacy of FMT. By gender, the calculated pooled rate of clinical success of FMT was 81% among males (95% confidence interval [CI]: 71%, 91%; *p* = 0.1; I^2^ = 40%) as opposed to an 84% success rate among their female counterparts (95% confidence interval [CI]: 78%, 90%; *p* = 0.4; I^2^ = 10%) as shown in Figure 3.

Efficacy outcomes in the patients with IBD who received FMT are tabulated in Table 6. Li, 2022 [2] and Aldrich, 2019 [4] did not differentiate the results for IBD patients. In Aldrich, 2019 [4], it was noted that 14% of the 175 subjects had known or were later diagnosed with IBD, the number of which was approximated to 25. In Nicholson, 2020 [5], out of the 120 IBD patients, only 111 had data available regarding FMT. In Barfield, 2018 [25], one patient required an additional delivery of FMT after 3 months from initial FMT due to the recurrence of symptoms and achieved resolution of CDI thereafter. In Brumbaugh, 2018 [26], one IBD patient required FMT twice for success while one patient still had persistent CDI despite also receiving FMT twice. In Russell, 2014 [30], one patient continued to experience gastrointestinal symptoms but remained negative on enzyme immunoassay (EIA) for CDI. The patient’s symptoms improved after initiating treatment for Crohn’s disease. That patient was counted as a success. There was one patient who was lost to follow up after 2 months and was counted as a failure. The overall success rate of FMT for CDI in IBD patients was 75.33% (223/296) while the failure rate was 24.66% (73/296). Although the total number of patients in the IBD cohort was 337, since there were two studies that did not report results for IBD patients, the numbers of patients from those studies were not included in calculating the overall success and failure rates.

### 3.2. Secondary Outcome

The secondary outcomes were serious adverse events (SAE) and adverse events (AE). Among the SAE and AE reported from the studies, only those that were determined to be attributable to FMT by the respective studies were included in our systematic review and meta-analysis. Most studies documented SAE and AE within 3 months after the administration of the FMT, except for the following: the follow-up periods of Aldrich, 2019 [4] and Barnes, 2018 [9] were 2 months and 2.5 months, respectively, while one patient in Russell, 2014 [30] was lost to follow-up after 4 months. 

There were 38 SAE in 36 patients and 47 AE in 45 patients as shown in Table 7. There was no death attributable to FMT. The causes of SAE were variable and there was no single predominant cause. While there were 20 IBD-related SAE in Nicholson, 2022 [1], SAE and AE were reported only for the IBD cohort and not for the non-IBD cohort. Among the AE, diarrhea and abdominal pain were commonly recorded.

The calculated pooled rate of serious adverse events was 2.0% (95% confidence interval [CI]: 0.0%, 3.0%; *p* = 0.1; I^2^ = 0.0%), and the pooled rate of adverse events was 15% (95% confidence interval [CI]: 5.0%, 25.0%; *p* = 0.02; I^2^ = 54.0% as shown in Figure 4. Based on the validation analysis (described below), Nicholson, 2022 [1] and Nicholson, 2020 [5] were discovered to have dominant effects on the analysis. Therefore, Nicholson, 2022 [1] was removed from analyses for both SAE and AE, while Nicholson, 2020 [5] was removed from the forest plot for AE (Figure 4).

### 3.3. Validation Analysis (Leave-One-Out Analysis)

To assess if any study involved in the main analysis had a dominant effect, a leave-one-out analysis was conducted. As shown in Table 8, upon removal of each study one by one, no significant impact on the summary statistics of the primary outcome or heterogeneity was found. However, in the analysis of the secondary outcomes, one study in both SAE and AE (Nicholson, 2022) and one in AE (Nicholson, 2020) had dominant effects on the summary statistics, which were excluded from the analysis. These studies were removed from forest plot analysis for SAE and AE.

## 4. Discussion

To our best knowledge, this is the first systematic review and meta-analysis of the safety and efficacy of FMT via all routes of administration in treating CDI and rCDI amongst pediatric patients. A prior systematic review by Iqbal et al., in 2018 demonstrated the effectiveness of encapsulated FMT to treat rCDI in patients of all ages [31]. The overall success rate from that systematic review was 92.6% (316/341); the overall success rate from our meta-analysis was 81.86% (740/904). The discrepancy in the efficacy can be explained by several factors. Firstly, Iqbal et al., included patients from all ages and there was only one study that consisted of data solely from pediatric patients [31,32]. Moreover, the review focused on FMT delivered via capsules only. On the other hand, our review focused solely on pediatric and adolescent patients up to and including the age of 21 and FMT administered via any route. These variables could have accounted for the difference in efficacy. Indeed, studies have concluded that encapsulated FMT may be considered in adult patients who previously failed to achieve resolution of CDI by other methods of FMT delivery such as colonoscopy [31,33,34]. A prior systematic review by Iqbal et al., concluded that encapsulated FMT was similar in efficacy and was associated with fewer adverse events when compared to colonoscopy [31]. In the same study, 8 out of 10 patients who previously failed FMT with colonoscopy achieved resolution of rCDI with encapsulated FMT [31]. Furthermore, routes such as colonoscopy, enema, nasogastric, or nasoduodenal tubes have been demonstrated to be inconvenient for patients [33], while oral preparation has been associated with greater ease of administration [34].

Our study confirmed a pooled success rate of 86% (95% CI: 77–95%, *p* < 0.001). Hence, FMT may still be used as an effective treatment for rCDI in pediatric patients. In fact, based on the success rate reported in the literature for the pediatric patient population, Infectious Diseases Society of America (IDSA) recommended, in 2017, that FMT may be considered in those who continued to experience multiple recurrences of CDI despite treatment with standard antibiotics [35,36].

However, despite its efficacy, FMT still remains relatively poorly regulated and standardized [36]. As discussed above, the routes of FMT administration may impact its effectiveness. Comorbidities and the gut microbiome composition of the recipient or the donor may also interfere with its therapeutic potential and may also be associated with adverse outcomes. In addition to the variabilities in success rate, since the delivery of FMT often requires a procedure such as colonoscopy, esophagogastroduodenoscopy, or use of enteral tubes, it also carries procedure-related risks [35]. The most common adverse outcomes from our review were diarrhea and abdominal pain or discomfort. The findings were consistent with the results previously reported in the literature for both adults and pediatrics [31,36]. Concerns have also been raised regarding SAE, such as transmission of multi-drug resistant organisms (MDRO), and blood-borne infections [35]. In 2019, the United States Food and Drug Administration released a safety alert and recommendations for more comprehensive screening and testing of the donors after two immunocompromised adult patients received FMT from a donor with positive extended-spectrum beta-lactamase (ESBL) producing Escherichia coli [36]. Both patients developed the infection and one subsequently expired. 

In our review, the rate of SAE and AE was low overall. The pooled rate for SAE was 2.0% (95% CI: 0.0–3.0%, *p* = 0.1) and the pooled rate for AE was 15% (95% CI: 5.0–25.0%, *p* = 0.02). The pooled rate for AE was slightly higher but comparable to the 12.5% that was derived from a systematic review of adult FMT patients by Iqbal et al. [31]. We also did not identify any deaths attributable to FMT, similar to the results by Iqbal et al., [31]. Furthermore, there was no infection caused by an MDRO after receiving FMT. Nonetheless, the causative pathogens from the aspiration pneumonia, appendicitis, or the infection unrelated to the gastrointestinal tract, which were part of SAE in our meta-analysis, were not known. It is also unclear whether the IBD-related hospitalizations or exacerbations were related to an infection episode. Moreover, viral RNA of SARS-CoV-2 has been found in the feces of infected patients [37]. Although there has not yet been documented transmission of COVID-19 via feces, the FDA recommended screening for the virus in potential donors [36]. Taking everything into consideration, we concur that donors should undergo a comprehensive screening process for infectious agents, including COVID-19, prior to the transplantation. While FMT has been successfully used to decolonize adult patients with MDROs, the available data is limited for similar use in pediatric patients and further research is warranted [38].

It is worth noting that both the recommendation from the FDA and the onset of the COVID-19 pandemic have led to changes not only in the practice of FMT administration but also the availability and access of FMT for pediatric patients [39]. For instance, the FDA also limited use of FMT to only emergent situations in 2020 and early 2021 during the peak of the COVID-19 pandemic [39]. A survey of pediatric gastroenterologists also showed that some practitioners took additional precautions such as increased screening of the donors or avoidance of FMT in immunocompromised patients. Some physicians increased the utilization of antibiotics such as fidaxomicin and oral vancomycin, while others entirely paused the program [39]. 

Dysbiosis has been established among IBD patients [40]. The success rate for FMT for CDI in pediatric patients with IBD from our review was 75.00% and was effective but was lower than the pooled success rate of 86.00%. The results were similar to prior observational studies such as Nicholson, 2022 [1], where the efficacy rates for overall, IBD, and non-IBD cohorts were 79.55, 75.68, and 81.85%, respectively. Nevertheless, Nicholson et al., concluded that there was no difference in FMT success between children with IBD and without IBD [1]. However, the authors discovered that a high proportion of children with FMT failure was found among those with clinically active IBD [1]. There are several confounding variables that can potentially influence the outcomes of FMT for CDI among IBD patients. Some examples include the type of stool received (e.g., fresh versus frozen stools), time from diagnosis of CDI until FMT, severity of CDI or IBD symptoms, mode of delivery, and concomitant medications, among others [1,40]. A systematic review and meta-analysis by Tariq et al. concluded that FMT is an effective therapy for rCDI in adult patients with IBD [41]. A similar analysis for pediatric patients has not been completed to the best of our knowledge.

Lastly, FMT has been associated with the restoration of the gut microbiome in CDI patients to the levels of healthy children [3]. Since the patients are at an age where the gut microbiome is still undergoing development at the time of receiving FMT, manipulation of the microbiome may influence metabolic or immune dysregulations [36]. In particular, restoration was noted in the levels of *Lachnospiraceae, Ruminococcaceae, Bifidobacteriaceae, Erysipelotrichaceae, and Bacteroidaceae*, the latter of which has been believed to be a key protective member of the gut microbiota [3]. Decolonization of *Enterobacteriaceae*, which can remain abundant in treatment-naive CDI patients prior to FMT, was also observed [3]. In other words, metagenomics may be a potential modality to predict long-term outcome of FMT in pediatric patients with rCDI. However, the presence of comorbidities such as IBD, immunocompromised status, or neurologic conditions may impede the process of reconstituting the gut microbiome and can result in the failure of FMT or requiring multiple administration of FMT [3]. The cost of metagenomic sequencing also presents a practical barrier to be performed in every patient receiving FMT. Thus, metagenomic sequencing of the gut microbiome may not be used as a predictor of long-term outcome in pediatric patients with complicated background comorbidities and/or in settings with limited resources [3].

### Limitations

We acknowledge several limitations. First, there were no randomized controlled trials in the previously published literature to include in our meta-analysis. Second, two studies needed to be excluded from the secondary outcome analysis to avoid a predominant effect on the analysis. Third, given that this is a systematic review and meta-analysis, we were not able to control for confounding variables in the patient qualities or the procedural protocols. Moreover, the determination of whether SAE or AE were attributable to FMT was made by the original authors. Since we were unable to review the original data of the studies, it is possible that the number of SAE or AE related to FMT may not accurately reflect the true number of adverse events. Fourth, there was evidence of publication bias present as shown by the Doi plot in Figure 5, Funnel plot in Figure 6, and an LFK index of greater than 4.

Lastly, we did not include Cho, 2019 [42] in our meta-analysis. The efficacy rate reported by Cho et al., was 75% from eight patients [42], which was comparable to that of the overall cohort of 904 patients (86%) and the IBD cohort of 296 patients (75.33%) from our meta-analysis. In addition, there was one SAE from Cho, 2019 [42] with abdominal pain and fever that was ultimately concluded to be related to influenza and not to FMT by the original authors. There were no other SAE or AE attributable to FMT from Cho, 2019 [42]. Therefore, while the study was not included in the meta-analysis, the impact was determined to be negligible in the current meta-analysis.

## 5. Conclusions and Future Directions

Despite its efficacy and safety profile, FMT remains a poorly regulated and standardized therapeutic modality. While there have been established data on FMT for adult patients with rCDI, similar information for the pediatric population has only emerged recently. Our systematic review and meta-analysis demonstrated that FMT can be an effective and safe therapy for pediatric and adolescent patients with rCDI. However, the effectiveness may be hindered by the presence of underlying conditions such as IBD or immunodeficiency. The availability and accessibility of FMT were also further deterred by the COVID-19 pandemic and the risk of transferring virulent pathogens during FMT. Although metagenomics provided promising data, there has yet to be an established tool to predict the long-term outcomes for the children who received FMT. There may also only be a few medical centers where such long-term monitoring can be provided. There is much to be explored regarding the use of FMT in the pediatric patient population, and robust data from randomized clinical trials is warranted.

## Figures and Tables

**Figure 1 microorganisms-10-02450-f001:**
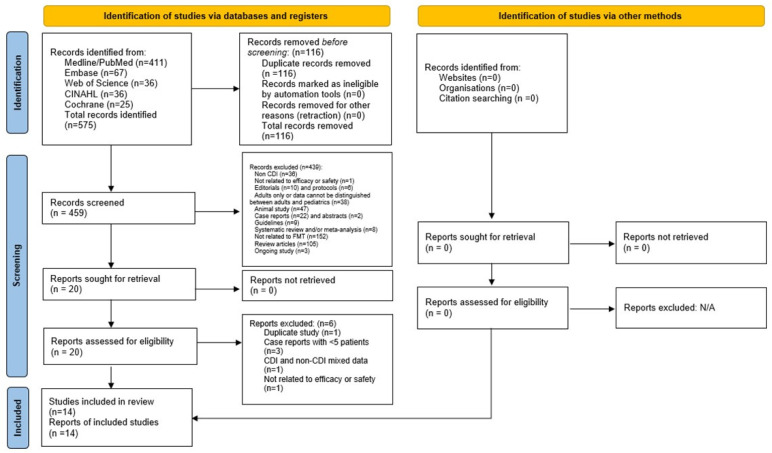
Preferred reporting items for systematic review and meta-analysis.

**Figure 2 microorganisms-10-02450-f002:**
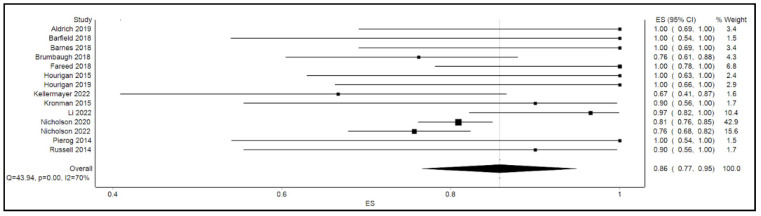
Forest plot displaying clinical success or efficacy of FMT.

**Figure 3 microorganisms-10-02450-f003:**
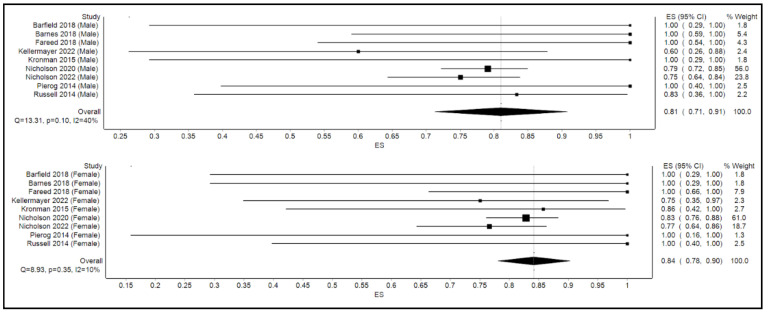
Forest plot displaying clinical efficacy of FMT by gender.

**Figure 4 microorganisms-10-02450-f004:**
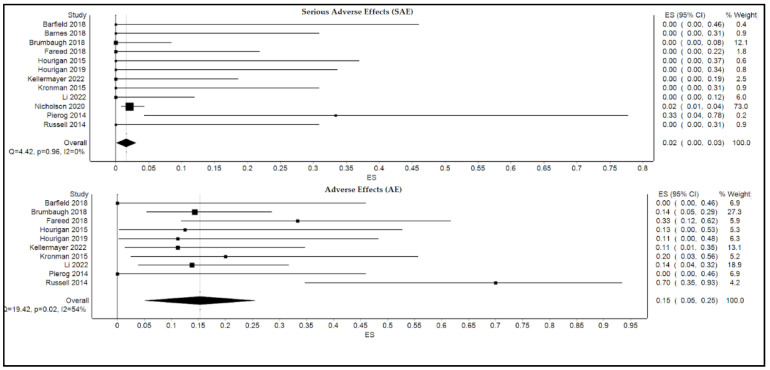
Forest plot displaying pooled rates of serious adverse events and adverse events of FMT.

**Figure 5 microorganisms-10-02450-f005:**
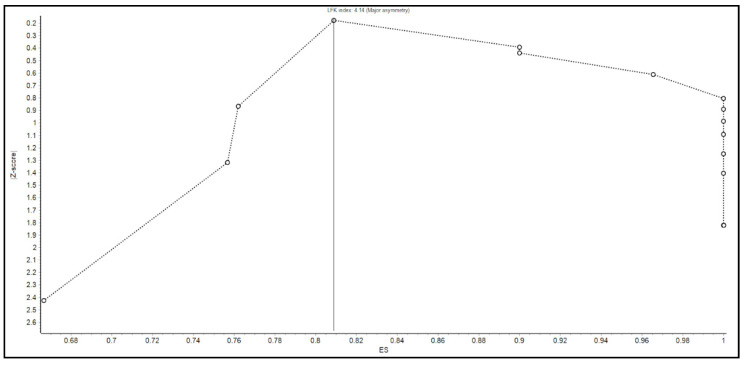
Doi plot for assessing the evidence of publication bias.

**Figure 6 microorganisms-10-02450-f006:**
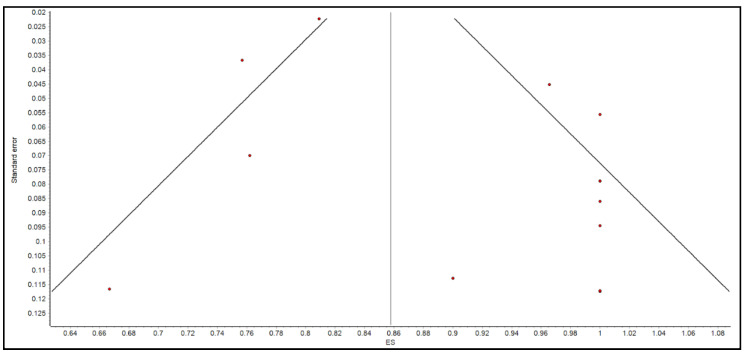
Funnel plot for assessing the evidence of publication bias.

**Table 1 microorganisms-10-02450-t001:** Summary of included studies.

Author/Year	Study Design	Quality Assessment	Score	Data Collection Period	Follow-Up Period (Months)	Number of Patients Who Received FMT	Single or Multi-Center	Condition Treated with FMT	FMT Method
Nicholson 2022 [1]	Retrospective	NOS	8	03/2012–03/2020	3	396	Multi-center	rCDI	Sigmoidoscopy/Colonoscopy Upper Gastrointestinal Delivery * Capsule
Li 2022 [2]	Retrospective	NOS	8	09/2014–09/2020	3	29	Single	rCDI	Naso-intestinal tube Enema Capsule
Kellermayer 2022 [3]	Prospective	NOS	7	02/2013–12/2015	2	18	Single	rCDI	Colonoscopy Enema Nasogastric tube
Aldrich 2019 [4]	Retrospective	NOS	9	01/2010–12/2014	2	10	Single	rCDI	Colonoscopy EGD Nasojejunal Gastric
Nicholson 2020 [5]	Retrospective	NOS	8	02/2004–02/2017	3	335	Multi-center	Both CDI and rCDI	Colonoscopy Enteral routes **
Hourigan 2019 [7]	Prospective	NOS	7	Not reported	6	9	Multi-center	rCDI	Colonoscopy
Barnes 2018 [9]	Prospective	NOS	8	06/2014–06/2016	2.5	10	Single	rCDI	Colonoscopy Upper Gastrointestinal Delivery ^&^
Fareed 2018 [10]	Prospective	NOS	8	Not reported	15	15	Multi-center	rCDI	Colonoscopy Nasojejunal
Barfield 2018 [25]	Case series	NIH scale	9	10/2013–11/2016	3	6	Single	Both CDI and rCDI	Colonoscopy
Brumbaugh 2018 [26]	Retrospective	NOS	7	03/2015–09/2016	3	42	Single	rCDI	Nasogastric or gastrostomy tube
Hourigan 2015 [27]	Prospective	NOS	7	Not reported	6	8	Multi-center	rCDI	Colonoscopy
Pierog 2014 [28]	Case series	NIH scale	7	Not reported	3	6	Single	rCDI	Colonoscopy
Kronman 2015 [29]	Case series	NIH scale	8	08/2011–05/2014	6	10	Single	rCDI	Nasogastric, nasojejunal or nasoduodenal tuve
Russell 2014 [30]	Case series	NIH scale	7	2009–2013	1 month to 48 months	10	Single	rCDI	Nasogastric tube Colonoscopy

NOS = Newcastle–Ottawa Scale; rCDI = recurrent *Clostridioides difficile* infection; CDI = *Clostridioides difficile* infection; EGD = esophagogastroduodenoscopy; * Upper delivery includes esophagogastroduodenoscopy, nasoduodenal or nasojejunal delivery. ** Enteral routes of administration include nasogastric or gastrostomy tube, nasoduodenal, nasojejunal, duodenal or jejunostomy tube. ^&^ Upper gastrointestinal delivery includes nasogastric, jejunal, or jejunal route.

**Table 2 microorganisms-10-02450-t002:** Routes of FMT administration.

Route of FMT Administration	Number of Times FMT Administered (*n* = 725)
Nasogastric tube	85 (11.72%)
Naso-intestinal tube	36 (4.97%)
Nasoduodenal	1 (0.14%)
Nasojejunal tube	6 (0.83%)
Gastric tube	1 (0.14%)
Capsule	31 (4.28%)
Enema	20 (2.76%)
Both esophagogastroduodenoscopy and colonoscopy	1 (0.14%)
Sigmoidoscopy	2 (0.28%)
Colonoscopy	361 (49.79%)
Unspecified route via upper gastrointestinal tract	63 (8.69%)
Unspecified route via lower gastrointestinal tract	105 (14.48%)
Unspecified upper or lower gastrointestinal tract	13 (1.79%)

FMT = fecal microbiota transplantation.

**Table 3 microorganisms-10-02450-t003:** Comorbidities found among the patients included in the systematic review and meta-analysis.

Comorbidities	Number of Patients
Inflammatory bowel disease (unspecified)	178
Ulcerative colitis	83
Crohn’s disease	76
Gastrointestinal diseases	49
Immunodeficient and/or transplant status	153
Malignancy	17
Neuromuscular disorders or impairment	8
Autism spectrum disorder	2
Genetic disorders	3

**Table 4 microorganisms-10-02450-t004:** Efficacy outcomes in the FMT cohort.

Author/Year	Number of Patients Who Received FMT	Number of Patients with FMT Success (Percentage)	Number of Patients with FMT Failure (Percentage)
Nicholson, 2022 [1]	396	315 (79.55%)	81 (20.46%)
Li, 2022 [2]	29	28 (96.55%)	1 (3.45%)
Kellermayer, 2022 [3]	18	12 (66.67%)	6 (33.33%)
Aldrich, 2019 [4]	10	10 (100%)	0 (0.00%)
Nicholson, 2020 [5]	335	271 (80.90%)	64 (19.10%)
Hourigan, 2019 [7]	9	9 (100%)	0 (0.00%)
Barnes, 2018 [9]	10	10 (100%)	0 (0.00%)
Fareed, 2018 [10]	15	15 (100%)	0 (0.00%)
Barfield, 2018 [25]	6	6 (100%)	0 (0.00%)
Brumbaugh, 2018 [26]	42	32 (76.19%)	10 (23.81%)
Hourigan, 2015 [27]	8	8 (100%)	0 (0.00%)
Pierog, 2014 [28]	6	6 (100%)	0 (0.00%)
Kronman, 2015 [29]	10	9 (90%)	1 (10.00%)
Russell, 2014 [30]	10	9 (90%)	1 (10.00%)
Total	904	740 (81.86%)	164 (18.14%)

FMT = fecal microbiota transplantation.

**Table 5 microorganisms-10-02450-t005:** Efficacy outcomes by gender.

Author/Year	Number of Male Patients with FMT Success	Number of Female Patients with FMT Success	Number of Male Patients with FMT Failure	Number of Female Patients with FMT Failure
Nicholson, 2022 [1] **	63	49	21	15
Li, 2022 [2]	NR	NR	NR	NR
Kellermayer, 2022 [3]	6	6	4	2
Aldrich, 2019 [4]	NR	NR	0 *	0 *
Nicholson, 2020 [5]	136	135	36	28
Hourigan, 2019 [7]	NR	NR	0 *	0 *
Barnes, 2018 [9]	7	3	0	0
Fareed, 2018 [10]	6	9	0	0
Barfield, 2018 [25]	3	3	0	0
Brumbaugh, 2018 [26]	NR	NR	NR	NR
Hourigan, 2015 [27]	NR	NR	0 *	0 *
Pierog, 2014 [28]	4	2	0	0
Kronman, 2015 [29]	3	6	0	1
Russell, 2014 [30]	5	4	1	0
Total	233 (78.98%; 233/295)	217 (82.51%; 217/263)	62 (21.02%; 62/295)	46 (17.49%; 46/263)

FMT = fecal microbiota transplantation; NR = not reported. * While the results were not reported by gender, Aldrich, 2019 [4], Hourigan, 2019 [7], and Hourigan, 2015 [27] reported 0 patients who failed FMT. ** Nicholson, 2022 [1] reported gender results only from the IBD cohort.

**Table 6 microorganisms-10-02450-t006:** Efficacy outcomes by IBD.

Author/Year	Total Number of IBD Patients Who Received FMT	Number of IBD Patients with FMT Success	Number of IBD Patients with FMT Failure
Nicholson, 2022 [1]	148	112	36
Li, 2022 [2]	16	NR	NR
Kellermayer, 2022 [3]	5	1	4
Aldrich, 2019 [4]	25	NR	NR
Nicholson, 2020 [5]	120 **	85	26
Hourigan, 2019 [7]	0	N/A	N/A
Barnes, 2018 [9]	0	N/A	N/A
Fareed, 2018 [10]	5	5	0
Barfield, 2018 [25]	2	2	0
Brumbaugh, 2018 [26]	13	7	6
Hourigan, 2015 [27]	5	5	0
Pierog, 2014 [28]	1	1	0
Kronman, 2015 [29]	3	3	0
Russell, 2014 [30]	3	2	1
Total	346	223 (75.33%; 223/296) *	73 (24.66%; 73/296) *

IBD = inflammatory bowel disease; FMT = fecal microbiota transplantation; NR = not reported; N/A = not applicable; * Since Li, 2022 [2] and Aldrich, 2019 [4] did not delineate outcomes for IBD patients, the number of patients from those studies were not included in calculating the rates of success or failure. ** Similarly, while there were 120 IBD patients who received FMT in Nicholson, 2020 [5], only 111 patients had their outcomes reported. Since the results of the remaining 9 patients were unknown, they were not included in assessing success or failure rates.

**Table 7 microorganisms-10-02450-t007:** Serious adverse events and adverse events.

Author/Year	Number of SAEs Related to FMT	Description of SAE	Number of Patients with SAE	Number of AE Related to FMT	Description of AE	Number of Patients with AE
Nicholson, 2022 [1]	29	19 IBD-related hospitalization; 1 pancreatitis-related hospitalization; 9 IBD-related surgeries	27	NR	NR	NR
Li 2022 [2]	0	N/A	0	4	1 fever; 1 transient diarrhea, 1 transient abdominal pain; 1 vomit	4
Kellermayer, 2022 [3]	0	N/A	0	2	1 paradoxical diarrhea 1 intermittent diarrhea and abdominal pain	2
Aldrich, 2019 [4]	NR	NR	NR	NR	NR	NR
Nicholson, 2020 [5]	7	1 aspiration pneumonia; 3 IBD flare; 2 colectomy; 1 vomiting and dehydration	7	19 *	Diarrhea Abdominal pain Vomiting	19
Hourigan, 2019 [7]	0	N/A	0	1	Chronic diarrhea and fecal urgency of non-infectious etiology	1
Barnes, 2018 [9]	0	N/A	0	NR	NR	NR
Fareed, 2018 [10]	0	N/A	0	5	5 abdominal pain	5
Barfield, 2018 [25]	0	N/A	0	0	N/A	0
Brumbaugh, 2018 [26]	0	N/A	0	6	6 vomiting	6
Hourigan, 2015 [27]	0	N/A	0	1	Diarrhea of non-infectious etiology	1
Pierog, 2014 [28]	2	1 appendicitis 1 infection unrelated to gastrointestinal tract	2	0	N/A	0
Kronman, 2015 [29]	0	N/A	0	2	1 vomiting 1 mucoid stools	1
Russell, 2014 [30]	0	N/A	0	7	1 mucoid stools; 3 abdominal pain and diarrhea; 1 diarrhea with abdominal pain; 2 bloody stools with bloating and abdominal pain	6
Total	38	N/A	36	47	N/A	45

IBD = inflammatory bowel disease; NR = not reported; N/A = not applicable; * Nicholson, 2020 [5] did not report the specific number of each AE.

**Table 8 microorganisms-10-02450-t008:** Outputs of sensitivity analysis (*n* = 14).

Studies	Pooled ES	LCI 95%	UCI 95%	Cochran Q	*p*	I^2^	I^2^ LCI 95%	I^2^ UCI 95%
Nicholson, 2022 [1]	0.876	0.777	0.976	34.999	0.000	65.713	38.297	80.947
Li, 2022 [2]	0.845	0.747	0.943	37.581	0.000	68.069	43.107	82.079
Kellermayer, 2022 [3]	0.861	0.769	0.952	41.218	0.000	70.887	48.797	83.447
Aldrich, 2019 [4]	0.853	0.760	0.946	40.570	0.000	70.422	47.863	83.220
Nicholson, 2020 [5]	0.894	0.811	0.978	35.512	0.000	66.208	39.312	81.184
Hourigan, 2019 [7]	0.853	0.760	0.947	41.123	0.000	70.820	48.662	83.414
Barnes, 2018 [9]	0.853	0.760	0.946	40.570	0.000	70.422	47.863	83.220
Fareed, 2018 [10]	0.847	0.755	0.939	36.915	0.000	67.493	41.935	81.801
Barfield, 2018 [25]	0.855	0.763	0.948	42.451	0.000	71.732	50.490	83.860
Brumbaugh, 2018 [26]	0.862	0.766	0.958	41.983	0.000	71.417	49.860	83.706
Hourigan, 2015 [27]	0.854	0.761	0.947	41.616	0.000	71.165	49.355	83.583
Pierog, 2014 [28]	0.855	0.763	0.948	42.444	0.000	71.728	50.481	83.858
Kronman, 2015 [29]	0.857	0.762	0.952	43.802	0.000	72.604	52.227	84.289
Russell, 2014 [30]	0.857	0.762	0.952	43.802	0.000	72.604	52.227	84.289

ES = effect size; LCI = lower confidence interval; UCI = upper confidence interval.

## Data Availability

Data supporting the statements can be found on PubMed, EMBASE, CINAHL, Web of Science, Cochrane and Google Scholar.

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
