# Peer review of "Efficacy and Safety of Fecal Microbiota Transplantation in Treatment of Clostridioides difficile Infection among Pediatric Patients: A Systematic Review and Meta-Analysis"

_microorganisms, 2022, doi:10.3390/microorganisms10122450_

Round 1

Reviewer 1 Report

The manuscript is extremely interesting and the meta-analysis work is described in detail. Delving  FMT and its applications ,especially in the pediatric setting, is a necessity considering  the paucity of investigations on this topic. In the last decade, FMT has boomed as a therapeutic option for the treatment of CDI and not only. Despite this, not all research groups in the world are keeping up with the latest and rapid updates in this regard. So, for this reason, I would suggest expanding the introduction section by adding:

-a short description on the operational protocols of preparing the fecal suspension for infusion;

- brief mention of the applications of fmt for extra intestinal disorders.

These two aspects could make the manuscript more complete and interesting.

Author Response

November 27th, 2022

To,

The Editor

Microorganisms

Dear Editors and Reviewers of Microorganisms,

We truly appreciate the opportunity to submit a revised version of our manuscript, titled, “Efficacy and Safety of Fecal Microbiota Transplantation in Treatment of Clostridioides difficile Infection Among Pediatric Patients: A Systematic Review and Meta-analysis” to your esteemed journal, Microorganisms.

Please see below our point-by-point response to the comments by the reviewers. We are uploading a clean and all changes accepted version as well as an underlined version of the revised manuscript that shows all the changes made to the original version of the manuscript.

We sincerely appreciate your kind consideration of this manuscript, and thank you for your time.

With Best Regards,

Kyaw Min Tun

Mark Hsu

Kavita Batra

Chun-Han Lo

Tooba Laeeq

Tahne Vongsavath

Salman Mohammed

Annie S. Hong

Reviewer #1’s comments

Comment #1: “I would suggest expanding the introduction section by adding:

  • A short description on the operational protocols of the preparing the fecal suspension for infusion;
  • Brief mention of the applications of FMT for extra intestinal disorders”

Response: Thank you for your suggestion. We absolutely agree that adding a short description of FMT preparation and brief mention of the FMT as treatment for extra-intestinal disorder will be insightful. We have made respective changes in lines 60-68 and lines 70-73 in the Introduction section.

Reviewer 2 Report

In the manuscript, Tun and colleagues present a systematic review regarding the safety and efficiency of FMT treatment in pediatric patients. Their work confirms the current trend in the field, showing that most studies were successful in reaching their objectives and concluding that FMT is both effective and safe at achieving either a remission (in case of IBD) or amelioration of symptoms (CDI). The topic is very important, as the potential of FMT has come into focus only in recent years. Moreover, the authors analysed pediatric studies which is even more needed given the fact that generally it is the adult group which receives the most attention in studies.

I appreciate the inclusion of the limitations chapter by the authors. Although not commonly seen in papers (usually only as a brief mention in the discussion), it serves to clarify that they clearly thought about the limiting factors which they could not account for. I think this manuscript meets the criteria set by the journal and I recommend it for publication. As I did not find any major issues, below are a few minor points I noticed:

- line 135: as the word "term" was written in quotation marks in line 132, I believe it is appropriate for the terms "failure" or "recurrence" to be written in quotation marks as well

- is it possible to enlarge either the entire figure 2, 3 and 4 (forest plots) or at least the text in them? In the current version of the manuscript, the text can be read only by manually zooming

- in the description of Table 6, there are three asterisks but in the table itself, there are only two (by the study by Nicholson et al., 2020), as the description with three asterisks is about Nicholson et al., 2020, I believe this to be a simple typographical error

- line 421: "Snce"

- lines 424-425: unless otherwise specified by the journal, names of taxa in latin are usually written in italic

- in Table 1, two different types of scales (in the "quality assessment" column, the NOS and NIH scale) are presented in the selected publications. What do the authors think about the comparability of those two scales, in other words, how accurately can the two scales be compared between each other?

Author Response

November 27th, 2022

To,

The Editor

Microorganisms

Dear Editors and Reviewers of Microorganisms,

We truly appreciate the opportunity to submit a revised version of our manuscript, titled, “Efficacy and Safety of Fecal Microbiota Transplantation in Treatment of Clostridioides difficile Infection Among Pediatric Patients: A Systematic Review and Meta-analysis” to your esteemed journal, Microorganisms.

Please see below our point-by-point response to the comments by the reviewers. We are uploading a clean and all changes accepted version as well as an underlined version of the revised manuscript that shows all the changes made to the original version of the manuscript.

We sincerely appreciate your kind consideration of this manuscript, and thank you for your time.

With Best Regards,

Kyaw Min Tun

Mark Hsu

Kavita Batra

Chun-Han Lo

Tooba Laeeq

Tahne Vongsavath

Salman Mohammed

Annie S. Hong

Reviewer #2’s comments

Comment #1: “Line 135: as the word “term” was written in quotation marks in line 132, I believe it is appropriate for the terms “failure” or “recurrence” to be written in quotation marks as well.

Response: Thank you for your comment. We agree with your suggestion and have added quotation marks to the words failure and recurrence.

Comment #2: “Is it possible to enlarge either the entire figure 2, 3, and 4 (forest plots) or at least the text in them? In the current version of the manuscript, the text can be read only by manually zooming.

Response: Thank you for your suggestion. We have enlarged the figures 2, 3, and 4 to the best of our abilities as you advised. However, attempts to further enlarge the figures will compromise the resolution of the figures and the text within them. Nevertheless, we did enlarge the figures as you suggested and hope that it improved the readability of the text.

Comment #3: “In the description of Table 6, there are three asterisks but in the table itself, there are only two (by the study by Nicholson et al., 2020) as the description with three asterisks is about Nicholson et al., 2020, I believe this to be a simple typographical error.”

Response: Thank you for kindly pointing out this error. It was indeed a typographical error and we have changed it to two asterisks.

Comment #4: “line 421: Since”

Response: Thank you for kindly pointing out this error. We have corrected it to “Since”.

Comment #5: “line 424-425: Unless otherwise specified by the journal, names of taxa in latin are usually written in italic.”

Response: Thank you for your suggestion. We have italicized the names of taxa.

Comment #6: “In Table 1, two different types of scales (in the “quality assessment” column, the NOS and NIH scale) are presented in the selected publications. What do the authors think about the comparability of those two scales, in other words, how accurately can the two scales be compared between each other?”

Response: Thank you for your comment. The NOS is typically used for observational studies such as prospective and retrospective studies. The NIH has different types of quality assessment tools available for different studies. We chose the NIH tool that is used for case series. Therefore, they are not comparable since they are used for different types of studies. Moreover, we also provided in the manuscript references that justify the use of each tool for their respective studies. We have provided additional references here as well:

NIH

  • American College of Cardiology/American Heart Association Task Force on Practice Guidelines, Obesity Expert Panel, 2013. Expert Panel Report: Guidelines (2013) for the management of overweight and obesity in adults. Obesity (Silver Spring). 2014;22 Suppl 2:S41-S410. doi:10.1002/oby.20660

NOS

  • Liu Y, Zhang Q, Jiang F, et al. Association between sleep disturbance and mental health of healthcare workers: A systematic review and meta-analysis. Front Psychiatry. 2022;13:919176. Published 2022 Jul 29. doi:10.3389/fpsyt.2022.919176
